# Impact of Coal Mining on the Moisture Movement in a Vadose Zone in Open-Pit Mine Areas

**Huiqin Lian [1], Haiyang Yi [2,*] , Yi Yang [1], Bin Wu [3] and Rui Wang [3]**

1   School of Safety Engineering, North China Institute of Science and Technology, Langfang 065201, China; lhuiq345@163.com (H.L.); yangyi19961103@163.com (Y.Y.)
2   School of Safety Supervision, North China Institute of Science and Technology, Langfang 065201, China
3   Heibei Coal Science Research Institute, Xingtai 054099, China; wu467060974@163.com (B.W.); wr19970621@163.com (R.W.)
*   Correspondence: yihaiyangchina@gmail.com

**Abstract:** Long-term dewatering of groundwater is a necessary operation for mining safety in open-pit coal mines, as extensive dewatering might cause ecological problems due to dramatic changes in moisture movement in the soil, especially in ecologically fragile areas. In order to evaluate the impact of the coal mining operation on moisture movement in the vadose zone and vegetation, this paper presents a quantitative methodology and takes the Baorixile open-pit coal mine as a study example. A long-term in situ experiment (from 2004 to 2018), laboratory analysis, and numerical modelling were conducted to analyze the mechanisms and relationship among the dropping groundwater level, the vadose-zone moisture, and the ecological responses in the grassland area. The experiment data and modelling results suggest that groundwater level dropping during open-pit mining operation has limited influence on the vadose zone, exhibiting a variation of capillary water zone within a depth of 3 m while the vadose zone and soil water zone were at least 16 m deep. The critical evaporation depth of ground water is 8 m. The long-term influence radius of groundwater dewatering is about 2.72 km during the Baorixile mining operation, and the groundwater level change mainly influences the lower part of the intermediate vadose zone and the capillary water zone below 16 m, with little influence on the moisture contents in the soil water zone where the roots of shallow vegetation grow. The results from this study provide useful insight for sustainable development of coal mining in ecologically fragile areas.

**Keywords:** open-pit coal mine; dewatering; groundwater level; vadose zone; moisture movement; capillary water

## 1. Introduction

Hulunbuir Grassland, the most concentrated and representative area of temperate grassland in China, includes multiple types of grassland ecosystems. It is not only an important ecological barrier in North and Northeast China [1], but also a main base for the integrated development of coal and electricity in China. Statistics suggest that Hulunbuir Grassland degrades at a rate of 2% annually and most of the rivers in Hulunbuir Grassland area have suffered from flow decline and even flow cutoff in recent years [2,3]. Eco-environmental problems have become a major risk, endangering the sustainable development of the mining area and even the ecological security of the entire region [4,5].

Environmental problems related to water in open-pit mines are increasingly piquing researcher's interests. In recent years, large efforts had been devoted to studying the consequence of water resources [6–12] and contamination migration [13,14] during open-pit coal mining. Furthermore, ecological issues due to dewatering in open-pit coal mining are also becoming attractive to researchers. As suggested by Chu et al. [15], the total ecological overburden of open-pit coal mines is 4.31 to 11.36 times that of underground coal mines. Therefore, great progress in studies has been made in terms of ecological

investigation [16–18], assessment [19–22] and management [23–26]. Corresponding theory models and techniques were also proposed in recent years [27–30], and the influencing mechanism was revealed to be the soil moisture change during water drainage in open-pit coal mining [22,31–33]. These works convinced us that soil moisture is indeed changing during open-pit coal mining processes. Thus, when concerning the impact of soil moisture on the plants near the open-pit coal mines, moisture movement in the vadose zone is a core factor.

Theories and study methods for moisture movement in vadose zones are relatively mature, such as Darcy's law, Richards' flow equation, and Philip's non-isothermal flow equation [34], etc. In recent years, progress had been made in the study of the principle of vadose zone moisture movement in different regions and under different environmental conditions [35,36]. Studies on the changes in moisture in vadose zones caused by coal mining are also showing an increasing tendency [37]. The majority of these studies can be divided into two categories: the movement and adsorption of pollutants in vadose zones, and the moisture movement in vadose zones under soil improvement. It is notable that these studies mainly focus on the influence of water bodies on vadose zone moisture during the process of surface infiltration at depths of less than 10 m (the depth of shallow vadose zones).

However, for thick and deep vadose zones below 10 m, research on the influence of deep groundwater level change on the moisture content of the upper vadose zone is quite limited. In the case of open-pit coal mining areas, the drainage and dewatering efforts of coal mines further aggravate the drop in groundwater level around the mining area. The water contained in vadose zones serves as an important link between groundwater and surface water and plays a vital role in the water circulation system. Because it disrupts the equilibrium of the initial distribution of soil moisture in a vadose zone, a drop in groundwater level poses a potential threat to the surface ecological environment.

Furthermore, in comparison with coal exploitation in the ecologically vulnerable area of the western grassland, research on the influence of high-intensity mining of open-pit mines in the eastern grassland region on the change in water resource and ecology is limited. The path through coal mining disturbs groundwater resources and the ecological environment can be summarized as "high-intensity mining of the coal, large-scale drainage of the stope, large drop in the groundwater level, change in the vadose zone moisture content, influence on the surface soil and vegetation growth, and then degradation of the ecological environment." The core issue is the influence mechanism and relationship among the coal mining, groundwater level, and change in moisture content in the vadose zone, which is also the basis for analyzing the influence of coal mining on the quality of the ecological environment.

The study methods commonly used on vadose zones in coal mine areas can be categorized as three types, namely, the system model, the conceptional model, and the water dynamic model [38]. The system model is mainly based on statistics theory [39], time sequence analysis [40], index regression [41], and the artificial neural network model [42]. The conceptional model involves the inflow and outflow of the vadose zone to express the moisture variation [43]. The water dynamic model is based on Richard's flow equation, which considers evaporation, groundwater level, and water transmission, etc. [44]. The system model is normally used to predict the tendency of moisture movement in a vadose zone, and the conceptional model is a widely used model in investigating large-scale areas that consider the water source of a vadose zone, but it is difficult to depict the moisture movement in a vadose zone [45]. The water dynamic model has limitations in modelling large-scale areas [46]. The present paper combines the conceptional model and the water dynamic model. The conceptional model was generated by large-scale field investigation, which enabled us to study the moisture in the vadose zone of a coal mine area quantitatively.

Therefore, this paper takes the Baorixile mine as an example, and aims at studying the impact of coal mining on the moisture movement in a vadose zone in open-pit mine

areas. Field investigation, laboratory tests, and numerical simulations were implemented to investigate the vadose zone moisture movement under the condition of a drop in groundwater level during mining and clarify the vertical distribution characteristics of vadose zone soil moisture, which are of great significance for correctly evaluating the influence of coal mining on regional groundwater resources and surface ecology, thereby guiding the management on the ecological policies and design of the proper way to rehabilitate.

## 2. Study Area and Methodology

### 2.1. General Characterization of the Study Region

The Baorixile open-pit mining area is located in Hailar District and Chenbaerhu Banner of Hulunbuir city, Inner Mongolia Autonomous Region (see Figure 1). The area has a continental sub-frigid zone climate, with an annual average temperature of −2.6 °C within the range of −48–37.7 °C, an annual average precipitation of 315.0 mm, and an annual average evaporation of 1344.8 mm. The Bao #1 mine has a production capacity of 20.0 Mt/year and has been in production for 13 years, since 2005. The mining area has a monoclinic structure, with a north-northwest dip direction and a 5–15° dip angle. Within the area, from bottom to top, the Lower Cretaceous Series of the Longjiang Formation, Damoguaihe Formation, and Yimin Formation, as well as the Quaternary System's Holocene series, are the mainly developed strata. The coal-bearing stratum is located in the Damoguaihe Formation of the Lower Cretaceous, where 20 minable coal seams in four minable coal seam groups have developed.

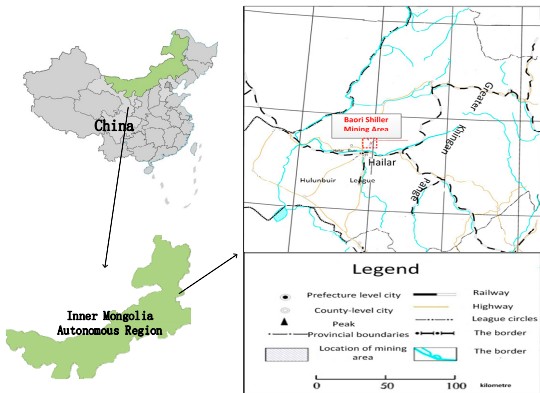

**Figure 1.** Geographical location map of the study area.

Within the mining area, the Quaternary porous water-bearing formation is a relatively stable aquifer affected by mining. The formation is of gravel stratum with a thickness of 17–30 m, which gradually becomes thinner from south to north and from west to east and wedges out to the east of the area. The Quaternary strata in the eastern part of the mining area is not water-bearing. The Cretaceous strata are directly covered by the Quaternary strata, the bottom of which is generally a layer of moraine pebbles, whereas the top of the coal-bearing strata is composed of mudstone, siltstone, and fine sandstone.

### 2.2. Evolution of Groundwater Level during the Mining Period

According to a report of a field investigation and water quality test by Inner Mongolia Coal Geological Exploration (Group) Co. Ltd. (www.nmmtdz.com.cn, accessed on 20 February 2021), from 2004 to 2016, the Baorixile open-pit mine recorded an annual average production capacity of 20 million tons/year, an annual average stripping capacity of 7.5 million m$^3$/year, and an annual average water discharge of 3 million m$^3$/year. In the early stage of dewatering, the water level elevation was 597.35 m, and the water discharged was 14,000 m$^3$/d. In 2004, the groundwater level elevation of the first mining area was 570 m, the dewatering capacity of which was 4900 m$^3$/d. The dewatering capacity of the stope was 8640 m$^3$/d in 2010, whereas the dewatering volume was 2941,100 m$^3$ in

2012. During the same period, affected by the dewatering and drainage of the pit, the groundwater level was negatively correlated with the amount of mining.

*2.3. Study Methodology*

Field investigation, laboratory tests, and numerical simulation were involved in studying the moisture movement in vadose zones. The numerical simulations were conducted to quantitatively describe the vertical movement of the moisture in vadose zones during groundwater level fluctuation.

## 3. Field Investigation

*3.1. Site Selection of the Field Investigation*

To elucidate the soil type of the vadose zone in the Baorixile mining area, 16 test points were positioned along the north-south and east-west directions based on the groundwater runoff direction (from northeast to southwest), with the cross-section intersection centered in the mining area. The test points were selected according to the degree of water resource utilization along the investigated directions. Then, in situ testing was conducted on the lithological and structural features of the vadose zone in the coalfield. For the locations of the test points, refer to Figure 2.

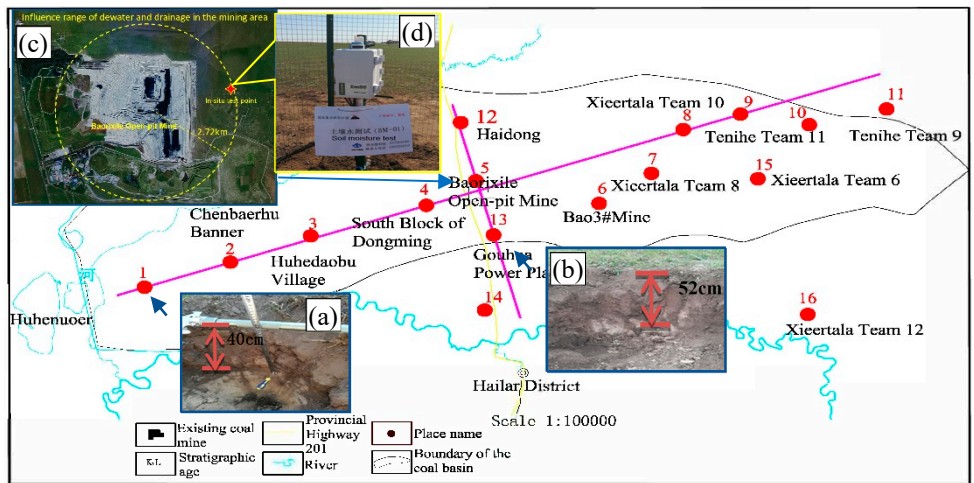

**Figure 2.** (**a**) Locations of field investigation of the vadose zone lithology, (**b**) pictures of on-site excavation, (**c**) satellite image of the field investigation on moisture, and (**d**) field measuring equipment.

*3.2. Results of Field Investigation*

3.2.1. The Lithologic Structure of the Vadose Zone

The on-site excavation and measurement (see Figure 2a,b) results show that the vadose zone is mainly composed of the upper humus layer and the lower deep sand soil layer, and there is a clear boundary between the two layers. The soil samples were analyzed for grain composition and then classified according to the international soil texture classification standard [47]. According to the data measured, the thickness of the humus layer is 30–60 cm, with an average thickness of approximately 40 cm. Comprehensive analysis suggested that the soil present at depths from 0 cm to 40 cm is a silt loam, and that below 40 cm is loamy sand.

3.2.2. Water Content Change along a Shallow Soil Profile

On 11 September, 2017, an in situ test field (N 49°24′50.35″, E 119°42′34.13″) (see Figure 2c) was constructed in the pasture near the dump site of the mining area. The test field included the selected typical profile, the humus soil thickness of which is 40 cm. Weather, soil moisture, and groundwater level data were obtained from 11 September, 2017, to 9 March, 2018 (180 days in total) (see Figure 2d). Meteorological elements were collected by an automatic weather station installed at the dump site. During the monitoring period,

the total precipitation was 1532.1 mm, the daily average net radiation was 4.68 MJ/m², the daily average temperature was −11 °C, the daily average relative air humidity was 68.7%, and the daily average wind velocity was 121.866 km/d.

The soil water content of the vadose zone was measured with EM − 50 m sensors. The soil water temperature and humidity sensor probes (type No. 5TM) were buried at 20 cm, 50 cm, 100 cm, and 300 cm below the ground. The data were collected every half an hour, and the water content changes obtained are shown in Figure 3.

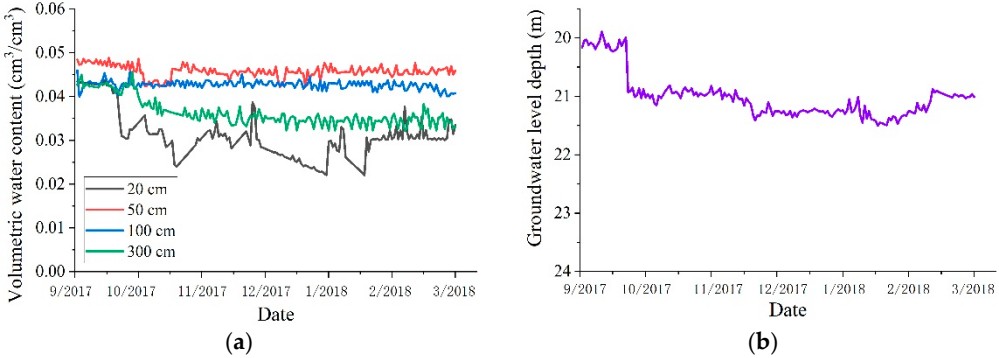

**Figure 3.** Evolution of (**a**) soil moisture content at different depths and (**b**) groundwater level during the monitoring period.

The moisture content of shallow soil at a depth of 20 cm varied greatly with time due to the low amount of precipitation and the dominant influence of evaporation during the monitoring period. The soil moisture content at a depth of 50 cm was the greatest and remained stable because of the decrease in soil texture and the effect of the local plants. The soil moisture content decreased with increasing depth from 100 cm to 300 cm, and the deeper the soil was, the more stable the water content. Generally, the soil moisture content increased with increasing depth. The groundwater level monitoring data were collected from the mine's groundwater level observation wells at a frequency of once a day. As it is plotted in Figure 3, from the establishment of the in situ test field to 3 October, 2017, the groundwater depth fluctuated around 20 m. Due to large-scale drainage activity at the Baorixile mine in early October, the groundwater level in the study area decreased. On 5 October, the groundwater level was 21 m, with a drop of 1 m. The groundwater level continued to decrease in December 2017 and fell to 21.5 m on 20 January, 2018; it rose again to 21 m in early March 2018.

## 4. Laboratory Tests

To quantitatively analyze the moisture movement in the vadose zone in the study area, measurement of the basic parameters of the vadose zone soil is required, including the soil bulk density, saturated hydraulic conductivity, and soil porosity. Equipment utilized in the laboratory tests is shown in Figure 4, including grading sieves, a variable head seepage device, and a nanoVoxel-3502E scanning platform for soil classification, hydraulic conductivity, and porosity, respectively.

### 4.1. Test Items

#### 4.1.1. Dry Soil Densities

The cutting-ring method was adopted to measure the soil bulk density of stratified sections [48]. The drying method was adopted to measure the stratified soil samples [49]. For each soil type after classification by grading sieves (see Figure 4a), four groups were measured and averaged.

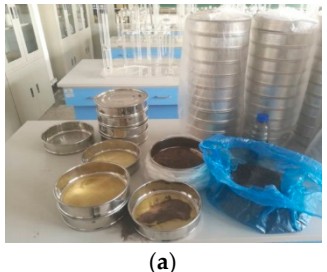 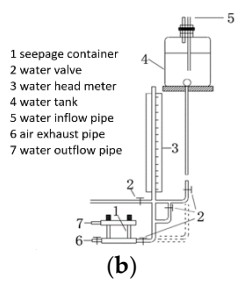 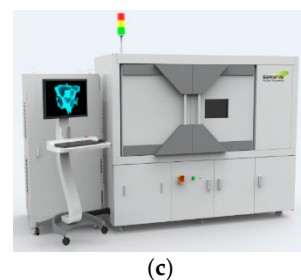

(**a**)        (**b**)        (**c**)

**Figure 4.** Equipment of laboratory tests on (**a**) soil classification, (**b**) hydraulic conductivity, and (**c**) porosity.

### 4.1.2. Saturated Hydraulic Conductivity

The variable head seepage method (see Figure 4b) was adopted [50] to measure the saturated hydraulic conductivity $k$ of the silt loam and loamy sand, and the average permeability coefficient $k_{20}$ of the two types of soil was calculated as follows:

$$k = 2.3 \frac{aL}{A(t_2 - t_1)} \log\left(\frac{h_1}{h_2}\right) \tag{1}$$

$$k_{20} = k_T \frac{\eta_T}{\eta_{20}} \tag{2}$$

where $a$ is the sectional area of the variable head tube (cm$^2$), $L$ is the height of the sample (cm), $A$ is the sectional area of the sample (cm$^2$), $t_1$ and $t_2$ are the starting time and the ending time of the head reading (s), $h_1$ and $h_2$ are the starting water head and the ending water head (cm), $\eta_T$ is the hydrodynamic viscosity coefficient (kPa·s) at T $^\circ$C, and $\eta_{20}$ is the dynamic viscosity coefficient (kPa·s) of water at 20 $^\circ$C. $\eta_T / \eta_{20}$, the dynamic viscosity coefficient ratio, is derived from related literature [51].

### 4.1.3. Porosity of Soil

To grasp the microstructure of the soil along a typical profile more comprehensively and precisely, a nanoVoxel-3502E scanning platform (see Figure 4c) and digital core analysis technology were used on the soil samples, 3D extraction of the soil pores was carried out with the threshold segmentation method, as plotted in Figure 5, and the equivalent pore diameters in the silt loam and loamy sand were determined.

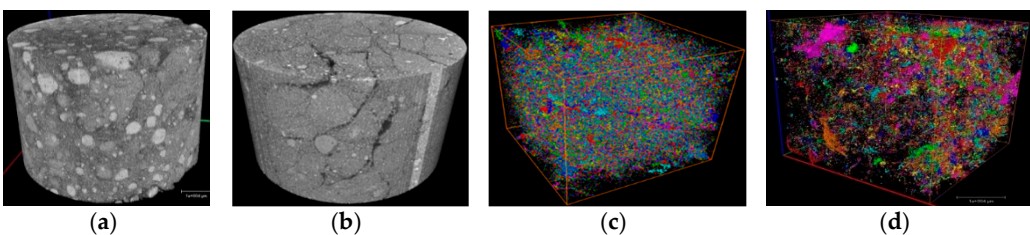

(**a**)        (**b**)        (**c**)        (**d**)

**Figure 5.** Scanning pictures of (**a**) silt loam and (**c**) loamy sand and 3D extraction of the pores and cracks in the (**b**) silt loam and (**d**) loamy sand.

### 4.2. Results of Laboratory Tests

The soil samples were analyzed for grain composition and then classified according to the international soil texture classification standard [47]. The soil present at depths from 0 cm to 0.4 m was a silt loam, which was the same as the field observation, and that below 0.4 m was loamy sand. The measured average dry soil density of silt loam was about 1.36 g/cm$^3$ and that of loamy sand was about 1.39 g/cm$^3$. According to the results obtained, the saturated moisture content of the silt loam was 0.4753 cm$^3$, and that of the loamy sand was 0.4214 cm$^3$. Based on the previously measured parameters of the soil profile, the Marqualdt–Levenberg parameter optimization algorithm in Hydrus was

used for the inversion [52] and to obtain the empirical parameter of the vadose zone's soil moisture characteristics. The inversion data were sourced from the moisture content data measured from 11 September, 2017, to 9 March, 2018 (180 days in total), based on which $\theta_s$ (cm$^3$/cm$^3$), the residual moisture content $\theta_r$ of the two layers, and the empirical parameter $\alpha$(cm$^{-1}$) were obtained. Specifically, Equation (3) was applied to determine the water retention curve [53]:

$$\theta = \theta_r + (\theta_s - \theta_r)/[1 + (\alpha h)^n]^m \tag{3}$$

which can be rewritten as

$$x_i = \ln(h_i), y_i = \ln\left\{[(\theta_s - \theta_r)/(\theta - \theta_r)]^{1/m} - 1\right\}, \beta = n \ln \alpha \tag{4}$$

Thus, Equation (4) can be simplified as

$$y = nx + \beta \tag{5}$$

In Equation (4), the subscript *i* means the $i_{\text{th}}$ number of measured data of $\theta_i$ and $h_i$. According to Equation (5), two steps are involved in the calculation [54]:

Step 1: the initial value of *m* is assumed to range from 0 to 1. Then calculate $\alpha$ with Equation (3).

Step 2: $m' = 1 - 1/n$. If $|m' - m| > \varepsilon$, where $\varepsilon$ is the accuracy error defined as 0.0001, return to step 1, otherwise complete the calculation.

All the calculations were implemented with Matlab software. The final characteristic parameters of the soil moisture are shown in Table 1.

**Table 1.** Characteristic parameters of soil moisture.

| Soil Depth /cm | Residual Moisture Content $\theta_r$/cm$^3$·cm$^{-3}$ | Saturated Moisture Content $\theta_s$/cm$^3$·cm$^{-3}$ | Empirical Parameter $\alpha$/cm$^{-1}$ | Curve Shape Parameter *n* | Hydraulic Conductivity $K_s$/cm·d$^{-1}$ |
|---|---|---|---|---|---|
| 0–40 | 0.043 | 0.4753 | 0.03028 | 1.39 | 9.623 |
| 40–2000 | 0.061 | 0.4214 | 0.005692 | 2.07 | 362.2 |

The test results of saturated transmissibility and average permeability of the study soils are listed in Table 2. The method for calculating the saturated water transmissibility and average permeability from laboratory test data is referenced in the literature [51,52]. The average permeability of silt loam was about 9.623 cm/d, and that of loamy sand was about 363.3 cm/d.

**Table 2.** Test results for saturated transmissibility and average permeability.

| | Time Variation (s) | Initial Water Head (cm) | Final Water Head (cm) | Saturated Water Conductivity at T °C $k_T$ (cm/s) | Correction $\eta_T/\eta_{20}$ | Saturated Water Transmissibility $k_{20}$ (cm/s) | Average Permeability $k_{20}$ (cm/s) |
|---|---|---|---|---|---|---|---|
| Silt loam | 8.15 | 100 | 73 | $1.08 \times 10^{-4}$ cm/s | 0.988 | $1.08 \times 10^{-4}$ cm/s | |
| | 7.63 | 100 | 71 | $0.94 \times 10^{-4}$ cm/s | 0.988 | $0.93 \times 10^{-4}$ cm/s | $1.11 \times 10^{-4}$ |
| | 8.99 | 80 | 67 | $1.18 \times 10^{-4}$ cm/s | 0.976 | $1.16 \times 10^{-4}$ cm/s | (=9.623 cm/d) |
| | 9.03 | 80 | 65 | $1.13 \times 10^{-4}$ cm/s | 0.988 | $1.12 \times 10^{-4}$ cm/s | |
| | 9.47 | 50 | 37 | $1.09 \times 10^{-4}$ cm/s | 1.012 | $1.09 \times 10^{-4}$ cm/s | |
| | 8.51 | 50 | 36 | $0.97 \times 10^{-4}$ cm/s | 0.988 | $0.96 \times 10^{-4}$ cm/s | |
| Loamy sand | 6.24 | 100 | 62 | $4.55 \times 10^{-6}$ cm/s | 0.976 | $4.53 \times 10^{-3}$ cm/s | |
| | 6.03 | 100 | 64 | $4.25 \times 10^{-6}$ cm/s | 0.988 | $4.25 \times 10^{-3}$ cm/s | $4.19 \times 10^{-3}$ |
| | 7.15 | 80 | 56 | $4.67 \times 10^{-6}$ cm/s | 0.976 | $4.66 \times 10^{-3}$ cm/s | (=362.2 cm/d) |
| | 6.22 | 80 | 55 | $4.19 \times 10^{-6}$ cm/s | 0.988 | $4.18 \times 10^{-3}$ cm/s | |
| | 6.35 | 50 | 29 | $4.26 \times 10^{-6}$ cm/s | 0.988 | $4.25 \times 10^{-3}$ cm/s | |
| | 5.89 | 50 | 27 | $4.63 \times 10^{-6}$ cm/s | 1 | $4.63 \times 10^{-3}$ cm/s | |

According to the test results of the water retention curves, hydraulic conductivity tests, saturated soil water content tests and numerical revision of Hydrus 1D, the parameters were summarized as in Table 1. The soil water retention curves obtained are shown in Figure 6a and the fitted curves by parameters in Table 1 are shown in Figure 6b.

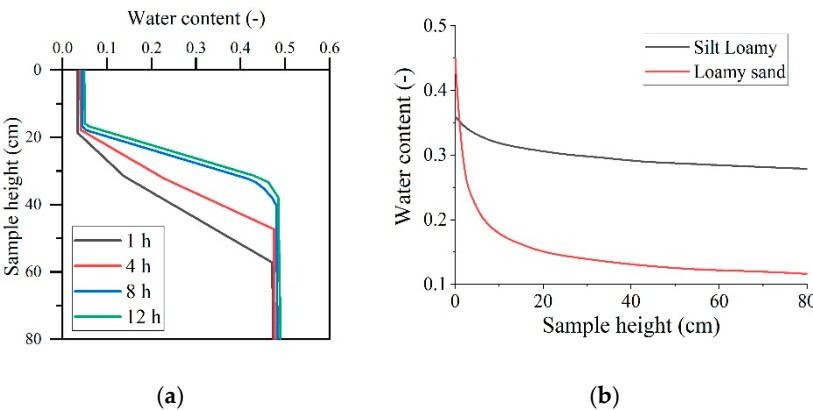

(**a**)　　　　　　　　　　　　　　　　(**b**)

**Figure 6.** Results of (**a**) water retention test of loamy sand and (**b**) the fitted curves of the soils.

The silt loam mainly contained fine pores with an equivalent diameter of 100 μm and had a porosity of 41.87%. The loamy sand contained pores with equivalent diameters of 100–500 μm and had a porosity of 49.36%.

## 5. Numerical Simulation

This paper involves numerical modelling of the moisture variation and evaporation discharge in the vadose zone induced by different groundwater levels. The current computational methods on these aspects are various, such as MODFLOW, STOMP, OpenGeoSys, TOUGH, and HYDRAUS [55–59]. Among them, HYDRAUS is a widely used software to calculate the vadose zone based on Richard's flow equation [60–62]. This paper applied HYDRUS software to compute the generalized model.

During coal mining, the influence radius of dewatering and drainage in the studied stope is approximately 2.72 km. According to the data monitored on site, the groundwater level near the boundary reaches 20–21.5 m, ranging within 1.5 m. These fluctuations are obvious and due to seasonal changes. With a thickness of approximately 20 m, the vadose zone in the mining area is deep.

To quantitatively study the influence of groundwater level changes on the moisture and movement in the vadose zone, the model was solved using the Hydrus-1D finite element calculation model, since the studied area was about 2.72 km, which was large enough to ignore the transverse water recharge, and as demonstrated by Miao et al. [26], moisture movement in a vadose zone is mainly triggered by evaporation. Thus, 1-D modelling is close to 3-D modelling if the main concern is on the vadose zone.

### 5.1. Generalized and Numerical Model Setup

According to the observation of on-site excavation and classification by grading sieves, the lithologic structure of the typical vadose zone profile was generalized into two layers within a total thickness of 20 m. The upper layer was a silt loam layer with a thickness of 0.4 m, the lower layer was a loamy sand layer with a thickness of 19.6 m, and each soil medium was homogeneous. Surface runoff was not considered into this study on the vadose zone in the study area during rainfall. Due to the low rainfall and strong evaporation in this grassland area, the moisture in the vadose zone is mainly exchanged vertically, and its lateral flow could be neglected. The generalized model is shown in Figure 7a.

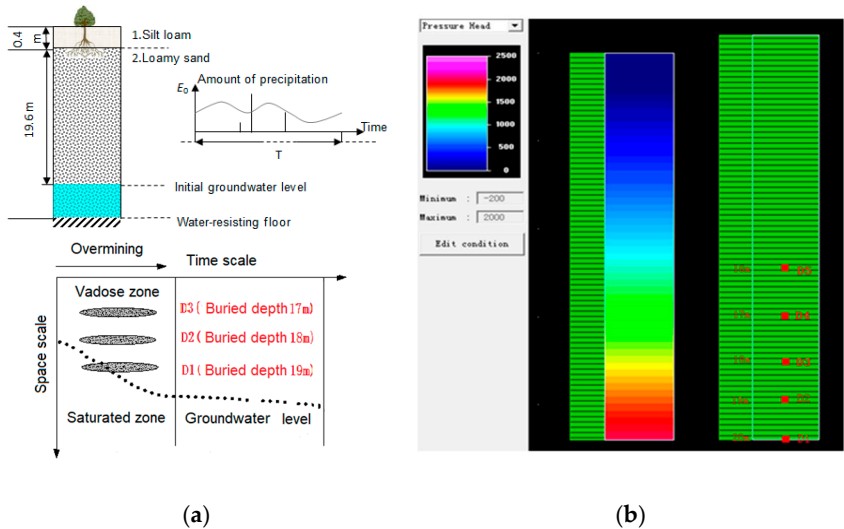

**Figure 7.** Schematic diagram of the (**a**) generalized model of the study area and (**b**) its numerical model.

The model was built based on the typical profile generalization results and parameters for calculations in two periods. The first period was the mining period (2004–2016), and the corresponding study was primarily focused on the changes in soil moisture content in the middle strip of the upper part of the capillary water belt, specifically the layer between the burial depths of 17 m and 19 m. The saturated moisture content can be taken as the Category I boundary of the lower boundary of the vertical motion of the soil moisture. The upper boundary condition was set as periodic rainfall. Three observation points were set at burial depths of 17 m, 18 m, and 19 m to monitor the changes in soil moisture content. The second period was the monitoring period (from 11 September, 2017, to 9 March, 2018), which was divided into three phases of 60 days each. The discretization setting of the soil profile was dissected into a total of 81 nodes at 25 cm equal intervals, and 5 observation points were set at burial depths of 16 m, 17 m, 18 m, 19 m, and 20 m (as shown in Figure 7b) to monitor the changes in soil moisture content.

### 5.2. Governing Equations

The Van-Genuchten model [63] was used to calculate the parameters of the soil moisture in unsaturated media, and the unsaturated permeability coefficient was predicted in the form of a water retention curve. The calculation method of the water retention curves was the same as that of references [50,64,65]. Specifically, it was assumed that the groundwater level at the bottom of the soil profile remained unchanged during a short period and that the water evaporation intensity of the soil surface was 0. After the moisture distribution in the soil reached a steady state, the moisture content data monitored at the depths of 20 cm, 50 cm, 100 cm, and 300 cm along the profile were fitted.

### 5.3. Definite Conditions

The initial conditions were set according to the field-measured moisture content. The upper boundary of model was set as the atmospheric boundary under natural conditions, which were mainly influenced by rainfall, evaporation, and crop transpiration, and irrigation was ignored in this problem. The amount of precipitation was directly measured by the automatic weather station in the area, and the crop transpiration was calculated with the help of the measured meteorological data and the Penman formula. The lower boundary was described with the pressure head indicated by the groundwater level. The fluctuation in the water level at an observation hole was used to represent the change in groundwater level and to select the deep drainage boundary. The bottom flux of the model was selected according to the fluctuation of the groundwater level.

## 6. Moisture Movement in a Vadose Zone during the Mining Period

In soil science, the vadose zone is divided into three belts from the surface to the water table: the soil water belt (capillary suspended water zone), the intermediate vadose belt, and the capillary water belt [66]. To compare the moisture movement and the range of capillary water rise under different groundwater extents, the variation in soil moisture content was chosen to reflect the vadose zone's moisture movement. The numerical simulation scheme was designed as follows:

(1) Based on years of weather data and groundwater level variation characteristics, the moisture movement in the typical profile of the vadose zone was simulated during the coal mining period (2004–2016). The soil moisture movement in the intermediate vadose belt, close to the capillary water belt, during the continuous lowering of the groundwater level was analyzed under the mining conditions.

(2) Based on the in situ test data, the moisture movement in the typical profile of the vadose zone during the monitoring period (from 11 September, 2017, to 9 March, 2018) was simulated. The influence of the groundwater level drop on the soil moisture contents in different vadose zone belts was analyzed, the maximum range of capillary rise was determined, and the critical depth to water, which can form a hydraulic relationship with the soil water belt, was simulated and predicted.

### 6.1. Groundwater Level and Vadose Zone Moisture Change during the Mining Period

Combined with the background of coal mining, the long-term moisture movement in the vadose zone under the conditions of periodic rainfall and a deep burial depth was simulated for the typical soil profile to obtain the influence of the long-term drop in the groundwater level on the moisture movement in different areas of the vadose zone.

The changes in soil moisture content at different burial depths during the mining period are shown in Figure 8. The moisture content of the entire vadose zone and its changes are closely related to the scale of coal mining in previous years. From 2004 to 2010, the groundwater level showed a downward trend, and the height of the capillary water belt and the content of soil moisture in the intermediate belt decreased. The groundwater level rebounded slightly only after the coal mining scale was decreased in 2014.

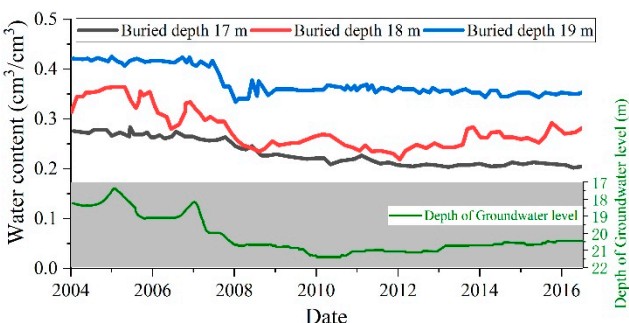

**Figure 8.** Changes in soil moisture content at different depths during the mining period.

Regarding the changes in groundwater level, the average annual yield of the Baorixile mine from 2004 to 2016 was 20 million tons. The initial water balance of the vadose zone was broken by the lowering of the groundwater level, which occurred at an average rate of 0.2 m/year. The change in soil moisture content with the lowering of the groundwater level were measured at three observation points with different burial depths: (1) At the burial depth of 17 m, the change in soil moisture content was small. When the groundwater level dropped by 3 m in 2010, the soil moisture content dropped from 0.28 cm$^3$/cm$^3$ to 0.25 cm$^3$/cm$^3$, a drop of 0.03 cm$^3$/cm$^3$. When the groundwater level dropped by 2.8 m in 2007, the soil moisture content at 1 m, 2 m, and 3 m above the capillary water belt decreased by 0.09 cm$^3$/cm$^3$, 0.06 cm$^3$/cm$^3$, and 0.03 cm$^3$/cm$^3$, respectively. (2) At the burial depth of 18 m, the soil moisture content decreased with a certain fluctuation, and

the maximum height of the capillary water belt was close to 2 m. When the groundwater level lowered by 2.8 m in 2007, the soil moisture content fluctuated between 0.3 $cm^3/cm^3$ and 0.35 $cm^3/cm^3$. When the groundwater level lowered by 3 m in 2010, the soil moisture content at this point was 0.28 $cm^3/cm^3$, i.e., a drop of 0.8 $cm^3/cm^3$. (3) At the depth of 19 m, when the groundwater level lowered by 2.8 m in 2007, the soil moisture content decreased from 0.43 $cm^3/cm^3$ to 0.34 $cm^3/cm^3$, and the maximum height of the capillary water belt dropped below 1 m, the largest drop recorded.

### 6.2. Groundwater Level and Vadose Zone Moisture Change during the Monitoring Period

The soil moisture contents of the profiles at the nodes (as shown in Table 3) were numerically calculated. The four-day meteorological data were used as the upper boundary input data, and the monitored value of the topsoil moisture content was used as the initial moisture content. The lower boundary conditions were set according to the depth changes in the groundwater level. The zonality of the vadose zone was determined, and the influence of the lowering of the groundwater level on the soil moisture content was analyzed according to the vertical distribution characteristics of the soil moisture.

**Table 3.** Comparison table for the groundwater depths and capillary water belt heights at different times.

| Time | 13 September 2017 | 5 October 2017 | 20 January 2018 | 1 March 2018 |
|---|---|---|---|---|
| Groundwater depths/m | 20 | 21 | 21.5 | 21 |
| Height of capillary water belt/m | 1.2 | 0.8 | 0.75 | 0.78 |

The simulation results are shown in Figure 9. Based on the changes in soil moisture content with depth, the ranges of the vadose zone's three belts (the soil moisture belt, the intermediate vadose belt, and the capillary water belt) were determined:

(1) As the soil moisture content started to decrease significantly at a depth of 4 m below the surface, the area from the surface to a depth of 4 m could be classified as the soil moisture belt. In this belt, the variation in soil moisture content decreased with the increasing depth, and moisture distribution along the depth in this belt was different, as in Figure 9. Specifically, from November to March, as the temperature decreased, as suggested by Shen et al. [67], the temperature of the shallow vadose zone decreased until the soil moisture was frozen, during which time the soil moisture content was significantly reduced and more stable. In March, the moisture content of the soil moisture belt increased significantly, mainly due to the melting of frozen soil and surface snow. The difference in the moisture within a depth of 4 m convinced us that it is easily influenced by atmospheric change.

(2) The area between the burial depths of 4–18.8 m was classified as the intermediate vadose belt. In this belt, the soil moisture content gradually decreased from 4 m to 9 m and remained basically stable from 10 m to 16 m, and its hydraulic connection with deep groundwater became weak. From the depth of 16 m to 19 m, the soil moisture content increased to a saturation state, making this area the capillary water belt. The soil moisture content was greatly affected by the variation in the capillary water belt's height. At a depth of 16 m, when the height of the capillary water belt was reduced by 0.4 m, the soil moisture content dropped from 0.08 $cm^3/cm^3$ to 0.06 $cm^3/cm^3$, indicating that the soil moisture close to the capillary water belt shifted downwards as a whole with the lowering of the groundwater level. The soil moisture content of the intermediate vadose belt within the depths of 4.2 m to 16 m was close to the residual water content of the soil and was basically stable. It can be inferred that the change in groundwater level had little effect on the moisture movement of the shallow intermediate vadose belt or the soil moisture belt above the depth of 16 m. This depth can be logically regarded as the critical depth at which the dewatering of coal mining can be influenced.

(3) The bottom of the vadose zone was a groundwater-supported capillary water belt with a soil moisture content of approximately 0.4 cm3/cm3. Affected by the drop in

groundwater level, the height of the capillary water belt varied from 0.75 m to 1.2 m. Specifically, on 13 September, 2017, the groundwater level reached 20 m, and the height of the capillary water belt was 1.2 m. After the groundwater level was reduced by 1 m due to large-scale dewatering and drainage in the mining area, the height of the capillary water belt dropped by 0.4 m to approximately 0.8 m. On 20 January, 2018, the groundwater level continued to drop to 21.5 m, but the height of the capillary water belt stabilized at 0.75 m, a drop of 0.05 m. On 1 March, 2018, the groundwater level returned to 21 m, and the height of the capillary water belt increased to 0.78 m, an increase of 0.03 m. The difference between the variation of capillary water belts was caused by the groundwater level, which led the soil water content to decrease with the increasing depth, thereby causing the decrease of moisture in the vadose zone along the capillary height.

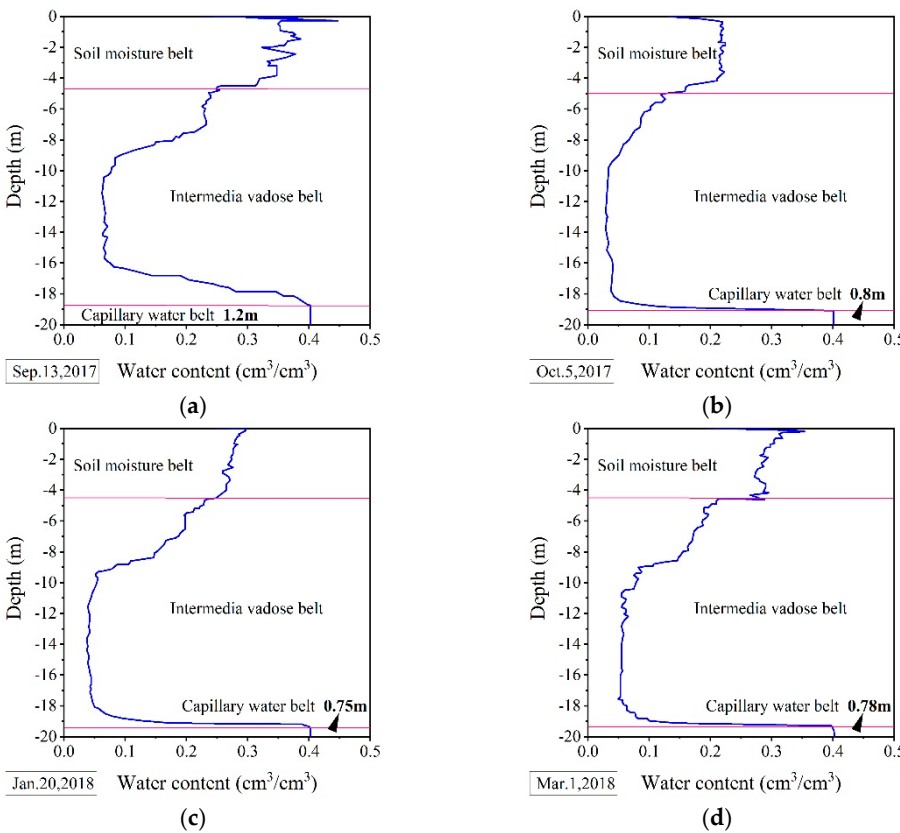

**Figure 9.** Simulated value of soil profile's moisture content on (**a**) 13 September 2017, (**b**) 5 October 2017, (**c**) 20 January 2018 and (**d**) 1 March 2018.

Based on the calculated results, the correlation curve between the vertical groundwater extent and the recharge amount is plotted in Figure 10. When the depth of the groundwater level was within 8 m, the recharge dropped from 421 mm/a to 100 mm/a and then remained unchanged, indicating that the recharge did not change as the groundwater level extended deeper than 8 m. Therefore, the critical evaporation depth of the typical profile of the vadose zone was 8 m. As the depth of groundwater level ranged from about 17 m to 21 m as displayed in Figure 8, according Figure 10, the groundwater level has little influence on the evaporation recharge. In other words, the fluctuation of groundwater level induced by dewatering and drainage in open-pit coal mines has limited impact on groundwater evaporation.

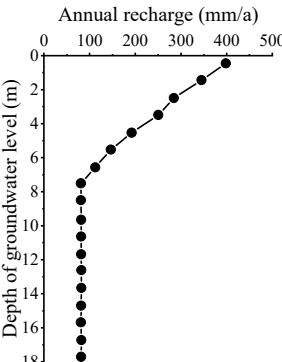

**Figure 10.** Variation in groundwater evaporation recharge with the groundwater depth.

## 7. Discussion

An investigation into the soil moisture content in Figures 8 and 9 convinced us that coal mining influenced the groundwater level initially but kept stable thereafter. This tendency can be also seen in the intermediate vadose belt. Dewatering and drainage in the coal mine caused approximately 0.4 m to 0.8 m change in depth of the capillary water belt. The critical evaporation depth of the study area was about 8 m. Based on these analyzed results, the impact of dewatering in coal mines on plants can be analyzed as follows.

Hu et al. [68] denoted that 10% of the effective diameter of soil is able to represent the diameter of the continuous capillary in soil. According to Zhang [67], the capillary rise height $h_c$ in soil can be defined as

$$0.73/D \leq h_c(cm) \leq 2/D \tag{6}$$

where D is the effective diameter of fine soil particles. As measured by Zhang [69], the average value of D in the Baorixile coal mine area is about 0.001 mm; thus, the variable height in the studied area ranged from 0.73 m to 2 m.

In the studied area, there were three kinds of plants, namely, grasses, shrubs, and trees. The depth of their roots buried were approximately 0.5 m, 1.5 m, and 6 m, respectively. Therefore, the critical depth that could be influenced by soil water content of grasses ranged from 1.23 m to 2.5 m, of shrubs ranged from 2.23 m to 3.5 m, and of trees ranged from 6.73 m to 8 m.

Obviously, the dewatering of coal mines has limited influence on the grasses and shrubs because the soil water content started to change below the depth of 4 m. Trees in the studied area would be influenced slightly due to the moisture in intermedial vadose zone caused by variation of underground water level, but would recover soon after the dewatering stops.

## 8. Conclusions

Through the investigation of the hydrogeology and lithological structure of the vadose zone in the Baorixile mining area, according to the analysis on the data of in situ tests and numerical modelling results, the following conclusions were drawn:

(1) The vadose zone in the Baorixile open-pit coal mine area has obvious zonality. The soil moisture belt was within a burial depth of around 4 m; the capillary water belt was between depths of about 0.75 m and approximately 1.2 m.

(2) When the groundwater depth was greater than 8 m, it was no longer hydraulically connected with the surface's soil moisture. The drop in groundwater level had a little effect on the moisture content of the vadose zone at depths of 16 m and above. The lower part of the intermediate vadose belt was significantly affected by the height variation in the capillary water belt. When the hydraulic connection between the soil and the water surface disappeared, the soil moisture content decreased by an average of about 15%.

(3) During the long-term variation in moisture content in the middle and lower parts of the deep vadose zone caused by the mining-induced drop in groundwater level, the groundwater level dropped by an average of about 0.2 m per year, the height of the capillary water belt decreased by 1 m, and the intermediate vadose belt affected by the capillary water belt was within 3 m of the surface.

(4) The groundwater level stabilized under long-term mining conditions and had little influence on the moisture movement in the vadose belt. In addition, the groundwater level change caused by the mining conditions mainly influenced the lower part of the intermediate vadose belt and the capillary water belt below the burial depth of 16 m but had little effect on the growth of shallow vegetation such as grasses and shrubs. Trees would be influenced initially but recovered to normal soon after the dewatering stops. As a whole, the vegetation was mainly affected by the changes in atmospheric precipitation at the upper boundary.

**Author Contributions:** Conceptualization, H.L. and H.Y.; methodology, Y.Y.; investigation, B.W. and R.W.; writing, H.Y.; visualization, Y.Y.; validation, H.Y. All authors have read and agreed to the published version of the manuscript.

**Funding:** This research was partially supported by the National Key R&D Program of China (2017YFC0804108) during the 13th Five-Year Plan Period, the National Science Foundation of China (51774136,52004090), the Natural Science Foundation of Hebei Province of China (D2017508099, E2020508025), the Program for an Innovative Research Team in the University sponsored by the Ministry of Education of China (IRT-17R37), the scientific program of the Education Department of Hebei Province (QN2019320) and the Fundamental Research Funds for the Central Universities (3142014018, 3142019019).

**Institutional Review Board Statement:** Not applicable.

**Informed Consent Statement:** Not applicable.

**Data Availability Statement:** Not applicable.

**Acknowledgments:** The authors of this paper thank the founding organizations for their financial effort. Thanks are also given to Prof. Zhenxue Dai for his help on the English editing herein.

**Conflicts of Interest:** The authors declare no conflict of interest.

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
