# Peer review of "Impact of Coal Mining on the Moisture Movement in a Vadose Zone in Open-Pit Mine Areas"

_sustainability, doi:10.3390/su13084125_

Round 1
Reviewer 1 Report
Overall, the manuscript presents an interesting topic, and it is well-written. However, the authors should recheck their text, as the verbal tense is not always coherent (the reasoning, especially in methodology, sometimes is in the present, sometimes is in the past – see, i.e., lines 145, 154).
The abstract follows the recommended guidelines and provides a general view of the developed work, although the authors fail to say the work objectively.
The introduction is clear and sound; however, it does not evidence the study's objective.
The description of the Study Area, in section 2.1, mentions the hydrological conditions, but few data are provided in this sense. I got the feeling that this section is a "general characterisation" of the region.
Section 2.2 provides "Evolution of groundwater level during the mining period". Where do the authors get this information from? I think it is pertinent to refer to the sources.
In the study scheme, it is referred that "16 test points are positioned along the north-south and east-west directions" (lines 131-132). How were these points chosen? What is the reasoning for this?
Lines 155-156, it is described that "probes are buried at 0 cm, 20 cm, 50 cm, 100 155 cm, and 300 cm" 0cm? Is that a mistake? If not, Figure 3 does not provide data regarding this deepness. Caption for Figure 3 should be placed on the same page as the figure (Line 160).
Line 184-185 says, "For each soil type, 4", here, number 4 should be written "four". Numeration up to 10 should always be written in full.
Lines 193-198 check the text formatting.
Line 202 "(Figs. 4)" does not follow the same formatting as the remain captions.
Lines 117-128 "The model is solved using the Hydrus-1D finite element calculation model, Since the studied area 217 is about 2.72 km, which is large enough to ignore the transverse water recharge" the word "Since" appears to have a misplaced capitalisation.
Line 249, "The Van-Genuchten model is used to calculate the parameters of the soil moisture in unsaturated" is missing the reference.
Lines 254-255, "According to the 254 water retention tests as that of reference of [41]", this "of" is with relation to…? Must be explained in the text (despite the reference).
Line 344, the caption of Figure 8, is on the wrong page.
Despite the fact that the article provides excellent insight into the developed work, I think it lacks organisation. Authors could improve the division between sections, evidencing the methodology and not mixing it with results and discussion. Additionally, what do the results mean in the light of what is already known? What do other authors say regarding this issue? As a reader, we get the sense to understand the pertinence of the study. However, how the methods are described makes it harder to follow the work.
Conclusions are coherent with the authors' results; however, they do not do more than that. This section should always answer the proposed objective (which is missing from the paper) and provide an insight into the real applicability of the developed work. In the abstract authors say that "The results from this study provide useful insight for sustainable development of coal mining in ecological-fragile areas." (Lines 26-27). Given the fact that this is a case study and that the authors do not discuss the real implications of their work (by using examples from other authors, for instance), I consider that this is a little bold.
Author Response
Firstly, thanks greatly for your great comments and suggestions on our paper manuscripts, which improved the quality of our paper to a large extent. We accept all your opinions and the corrected points have been marked by red color or track formation.
Thanks again for your great efforts on prompting our paper.
Reply to #REVIEWER 1
Overall, the manuscript presents an interesting topic, and it is well-written. However, the authors should recheck their text, as the verbal tense is not always coherent (the reasoning, especially in methodology, sometimes is in the present, sometimes is in the past – see, i.e., lines 145, 154).
Re: Thanks, We have checked all the text on verbal tense. Please see the revised words such as lines 42, 57, 182, 190 to 202, 245 to 256 etc.
The abstract follows the recommended guidelines and provides a general view of the developed work, although the authors fail to say the work objectively.
Re: Thanks, the objective of this study work is to evaluate the impact of coal mining operation on moisture movement in vadose zone and vegetation. Which have been clarified in line 13 to 14 of abstract.
The introduction is clear and sound; however, it does not evidence the study's objective.
Re: Thanks, we have modified the introduction and clear the study objective in line 99 to 102.
The description of the Study Area, in section 2.1, mentions the hydrological conditions, but few data are provided in this sense. I got the feeling that this section is a "general characterisation" of the region.
Re: Thanks, We accept. The title of 2.1 has been corrected as general characterisation of the study area. Please see line 109.
Section 2.2 provides "Evolution of groundwater level during the mining period". Where do the authors get this information from? I think it is pertinent to refer to the sources.
Re: Thanks. The data resource comes from the test results by Inner Mongolia coal geological exploration (Group) Co. Ltd. Which has been referenced in line 132 and 133.
In the study scheme, it is referred that "16 test points are positioned along the north-south and east-west directions" (lines 131-132). How were these points chosen? What is the reasoning for this?
Re: Thanks, the reason for the points selection is based on the degree of water resource utilization along the investigate directions, which has been clarified in line 153-154.
Lines 155-156, it is described that "probes are buried at 0 cm, 20 cm, 50 cm, 100 155 cm, and 300 cm" 0cm? Is that a mistake? If not, Figure 3 does not provide data regarding this deepness. Caption for Figure 3 should be placed on the same page as the figure (Line 160).
Re: Thanks, indeed, it is a mistake and it has been corrected, please see line 183. The caption for Figure 3 has been placed on the same page as the figure.
Line 184-185 says, "For each soil type, 4", here, number 4 should be written "four". Numeration up to 10 should always be written in full.
Re: Thanks, it has been correceted, please see line 219.
Lines 193-198 check the text formatting.
Re: Thanks, it is cause by the formulation of mathtype, we have asked help from editors.
Line 202 "(Figs. 4)" does not follow the same formatting as the remain captions.
Re: Thanks, we have checked all the citation of figures in text.
Lines 117-128 "The model is solved using the Hydrus-1D finite element calculation model, Since the studied area 217 is about 2.72 km, which is large enough to ignore the transverse water recharge" the word "Since" appears to have a misplaced capitalisation.
Re: Thanks, it has been corrected, please see line 307.
Line 249, "The Van-Genuchten model is used to calculate the parameters of the soil moisture in unsaturated" is missing the reference.
Re: Thanks, the missed reference has been added to line 338.
Lines 254-255, "According to the 254 water retention tests as that of reference of [41]", this "of" is with relation to…? Must be explained in the text (despite the reference).
Re: Thanks, it has been corrected and explained in line 340.
Line 344, the caption of Figure 8, is on the wrong page.
Re: Thanks, it has been corrected to keep the caption and figure on the same page.
Despite the fact that the article provides excellent insight into the developed work, I think it lacks organisation. Authors could improve the division between sections, evidencing the methodology and not mixing it with results and discussion. Additionally, what do the results mean in the light of what is already known? What do other authors say regarding this issue? As a reader, we get the sense to understand the pertinence of the study. However, how the methods are described makes it harder to follow the work.
Re: Thanks for your comment, we accept and improved the division between sections. Please see the revise paper manuscript, the study methodologies and results are introduced separately.This paper aims at evaluate the impact of dewatering of open-pit mine on the moisture in vadose, which maybe the source of ecological issues exists in the grassland areas. We have added a section of discussion in the paper to analysis this purpose.
Conclusions are coherent with the authors' results; however, they do not do more than that. This section should always answer the proposed objective (which is missing from the paper) and provide an insight into the real applicability of the developed work. In the abstract authors say that "The results from this study provide useful insight for sustainable development of coal mining in ecological-fragile areas." (Lines 26-27). Given the fact that this is a case study and that the authors do not discuss the real implications of their work (by using examples from other authors, for instance), I consider that this is a little bold.
Re: Thanks greatly for your comment, frankly, this paper is a primary work of our current study, our study works in this paper is clear whether the operation of open-pit coal mine cause problems on the vegetations by underground water level. In recent years, the ecological problem is the study area is becoming fragile, but the reason is still not clear. In our work, we take a typical coal mine, the Baorixile open-pit mine, which is largest coal mine in Hulunbuir grassland, as a study case.
Reviewer 2 Report
Section 3.2.2
Unfortunately, there are no examples of calculating saturated transmissibility and average permeability coefficient. In this regard, there is no way to evaluate the calculations of these parameters.
Раздел 3.3.1
For the first time in this paragraph, it is said that the thickness of the loamy sands is 20 m. Homogeneous layer of 20 meters is impossible. What method was used to study the soil section?
The statement "the presence of a layer of silty loam accelerates the penetration of water after rain" is very controversial and depends on many factors.
Section 3.3.4
There are no examples of calculating characteristic Parameters of Soil Moisture in the table 1.
Section 4.2
The scale of the axes of Figure 9 is not chosen very well. Conclusions from Figure 9 are not clear.
Author Response
Firstly, thanks greatly for your great comments and suggestions on our paper manuscripts, which improved the quality of our paper to a large extent. We accept all your opinions and the corrected points have been marked by red color or track formation.
Thanks again for your great efforts on prompting our paper.
Reply to #REVIEWER 2
Unfortunately, there are no examples of calculating saturated transmissibility and average permeability coefficient. In this regard, there is no way to evaluate the calculations of these parameters.
Re: Thanks, we have listed the test data of saturated transmissibility and average permeability in the text, please see Table 2.
Раздел 3.3.1
For the first time in this paragraph, it is said that the thickness of the loamy sands is 20 m. Homogeneous layer of 20 meters is impossible. What method was used to study the soil section?
Re: Thanks, it has been corrected as two layers existed within a total thickness of 20 m. please see line 270. The soil sections were studied by grain composition and classification according to texture classification standard, which has been announced in line 312 to 313 in the revised version.
The statement "the presence of a layer of silty loam accelerates the penetration of water after rain" is very controversial and depends on many factors.
Re: Thanks, We accept. this sentence has been removed out from revised manuscript, please see line 316.
Section 3.3.4
There are no examples of calculating characteristic Parameters of Soil Moisture in the table 1.
Re: Thanks, the calculation method were introduced in line 257 to 270.
Section 4.2
The scale of the axes of Figure 9 is not chosen very well. Conclusions from Figure 9 are not clear.
Re: Thanks, we have corrected the axes title of Figure 9. And additional analysis on this figure has been conducted in line 464 to 467 and 475 to 479.
Reviewer 3 Report
The problem of rock mass drainage caused by mining activities (both underground and open pit) is an important and current problem. It affects many areas of the world. I agree with the authors that it has grown in importance in recent years. The results of the observations are interesting and constitute good material for analysis. The numerical calculations performed are correct. The parameters of the computational model were selected in an appropriate manner. The article is written correctly and it is interesting. It’s a valuable source of information for people dealing with this field in practice.
Nevertheless, I have two comments on the article. Some conclusions as well as the topic suggest that the article is universal. On the other hand, the presented content, observation results, modeling results, parameters, mining and geological conditions, etc., relate only to one specific case - Baorixile open-pit coal mine. No general conclusions can be drawn in the case of such a specific case of exploitation. The second issue is the problem of the literature used. As the authors themselves admit, the problem of rock mass drainage is important and topical on a global scale. There are many significant achievements in this field that could also be applied in this case (e.g. computational methods).
Nevertheless, after taking these changes into account, in my opinion, the article will be a valuable source of information. The level of work and the subject matter of the article correspond to the requirements of the journal.
Author Response
Firstly, thanks greatly for your great comments and suggestions on our paper manuscripts, which improved the quality of our paper to a large extent. We accept all your opinions and the corrected points have been marked by red color or track formation.
Thanks again for your great efforts on prompting our paper.
Reply to #REVIEWER 3
The problem of rock mass drainage caused by mining activities (both underground and open pit) is an important and current problem. It affects many areas of the world. I agree with the authors that it has grown in importance in recent years. The results of the observations are interesting and constitute good material for analysis. The numerical calculations performed are correct. The parameters of the computational model were selected in an appropriate manner. The article is written correctly and it is interesting. It’s a valuable source of information for people dealing with this field in practice.
Re: Thanks greatly for your comments on our study works.
Nevertheless, I have two comments on the article. Some conclusions as well as the topic suggest that the article is universal. On the other hand, the presented content, observation results, modeling results, parameters, mining and geological conditions, etc., relate only to one specific case - Baorixile open-pit coal mine. No general conclusions can be drawn in the case of such a specific case of exploitation. The second issue is the problem of the literature used. As the authors themselves admit, the problem of rock mass drainage is important and topical on a global scale. There are many significant achievements in this field that could also be applied in this case (e.g. computational methods).
Re: Thanks, we accept and the paper has been modified. For the first comment, we have discussed the previous results in a separated section. We took the largest open-pit coal mine in Hulunbuir grassland as a study case, which has representative characters in the study area, of course, this has limits, we are doing further study on this area. Secondly, the study methods, in terms of laboratory test and numerical modelling referenced from literatures have been cited in the text, please see lines 246, 257, 265, 276 and 34, 294 to 298.
Nevertheless, after taking these changes into account, in my opinion, the article will be a valuable source of information. The level of work and the subject matter of the article correspond to the requirements of the journal.
Re: Thanks greatly for your comments on improving our paper.
Reviewer 4 Report
Impact of Coal Mining on the Moisture Movement in a Vadose Zone in Open-pit Mine Areas
- The aim of this paper has been mentioned as “This paper presents a quantitative methodology to evaluate the impact of the coal mining operation on moisture movement in the vadose zone by taking the Baorixile open-pit coal mine as an example.” Is there any other technique available currently? Please add a paragraph to the Introduction section, explain the current available methods, and mention why this new proposed technique is useful and what are the advantages of this technique over other methods? If no method is currently available, please also mention.
- Line 51, add space between coal mining and the reference [22,3-33].
- Line 66, delete “with”: However, for thick and deep vadose zones with below 10 m, research on the influence of deep groundwater level change on the moisture content of the upper vadose zone is quite limited.
- Line 138: Update Figs. 2 to Figure 2 to keep consistency in the manuscript, also line 202, change Figs. 4 to Figure 4.
- Improve the quality of Figure 2.
- Line 157 and 160, remove excessive space between Figure and 3.
- Use same format for reporting figures through the manuscript:
Figure 2. Distribution of the vadose zone lithology test points
Figure 3. Evolution of (a) Soil Moisture Content at Different Depths and (b) groundwater level 160 during the Monitoring Period.
- Lines 208-209: It is important to address the pictures in order; however, you have addressed picture (a) first and then (c), (b) and (d). Please reorder.
- It is required to add space between line 205 and Figure 4.
- Line 217: To quantitatively study the influence of groundwater level changes on the moisture The model is solved using the Hydrus-1D finite element calculation model, Ssince the studied area is about 2.72 km, which is large enough to ignore the transverse water recharge, and as demonstrated by Mao et al. [26], moisture movement in vadose zone is mainly triggered by evaporation, thus 1-D modelling is close to 3-D modelling if the main concern is on the vadose zone. Also please try to divide this sentence in to two.
- Line 234, add space between Figure and 5.
- Line 309, update Figure .7 to Figure 7.
- Line 326, Add “was”: The largest drop was
- Line 327, Delete repeated word: 4.2. Groundwater Groundwater Level and Vadose Zone Moisture Change during the Monitoring Period
- Please modify Table 2 to have 2017.9.13 in one line.
- Line 337: Add a space between Table 2 and line 337.
- Line 337: Write Figure 8 instead of Figs. 8
- Authors are required to provide better comparison between the results from Figure 8 to point out the differences between different conditions and the reason for that.
- Line 344: add space between (a) and Sep. 13 and also add (d) for Mar. 1, 2018.
- From line 345 to conclusion: a series of discussion has been provided without explaining where they belong. Are they the discussion for the total project? Please specify.
- Lines 346-348: please rewrite the sentence.
- Line 368: Update cm3/cm3 to cm3/cm3 to keep consistency in the manuscript
- Line 391: please add “a” to The drop in groundwater level has a little effect on the moisture content of the vadose zone at depths of 16 m and above. Same comment for line 404.
- Is there any verification for the numerical model? Have the authors compared the results from numerical simulation with the in-situ results? The study focus on an important area but there is a lack of connection between different sections which required to be improved.
- Is there any recommendation to be provided by the authors? Did the model work successfully?
- The conclusion needs to be improved as it only covers the summary of findings without provide a proper conclusion.
Author Response
Firstly, thanks greatly for your great comments and suggestions on our paper manuscripts, which improved the quality of our paper to a large extent. We accept all your opinions and the corrected points have been marked by red color or track formation.
Thanks again for your great efforts on prompting our paper.
Response to #REVIEWER 4
- The aim of this paper has been mentioned as “This paper presents a quantitative methodology to evaluate the impact of the coal mining operation on moisture movement in the vadose zone by taking the Baorixile open-pit coal mine as an example.” Is there any other technique available currently? Please add a paragraph to the Introduction section, explain the current available methods, and mention why this new proposed technique is useful and what are the advantages of this technique over other methods? If no method is currently available, please also mention.
Re: Thanks greatly for your comments on our paper manuscript. We have added a separated paragraph to the introduction to explain current available methods and clarified the advantage of our methods. Please see line 85 to line 98
- Line 51, add space between coal mining and the reference [22,3-33].
Re: Thanks, we have corrected it. Please see line 51.
- Line 66, delete “with”: However, for thick and deep vadose zones with below 10 m, research on the influence of deep groundwater level change on the moisture content of the upper vadose zone is quite limited.
Re: Thanks, we have corrected it. Please see line 66.
- Line 138: Update Figs. 2 to Figure 2 to keep consistency in the manuscript, also line 202, change Figs. 4 to Figure 4.
Re: Thanks, we have corrected it. Please see the line 159 and line 214.
- Improve the quality of Figure 2.
Re: Thanks, we have improved the quality of Figure 2, please see Figure 2 in Page 4.
- Line 157 and 160, remove excessive space between Figure and 3.
Re: Thanks, we have corrected it. Please see line 188.
- Use same format for reporting figures through the manuscript:
Figure 2. Distribution of the vadose zone lithology test points
Figure 3. Evolution of (a) Soil Moisture Content at Different Depths and (b) groundwater level 160 during the Monitoring Period.
Re: Thanks, we have unified the format of figures, please see the caption of Figure 2.
- Lines 208-209: It is important to address the pictures in order; however, you have addressed picture (a) first and then (c), (b) and (d). Please reorder.
Re: Thanks, we have reordered the pictures, please see Figure 5.
- It is required to add space between line 205 and Figure 4.
Re: Thanks, we have corrected it. Please see line 211.
- Line 217: To quantitatively study the influence of groundwater level changes on the moisture The model is solved using the Hydrus-1D finite element calculation model, Ssince the studied area is about 2.72 km, which is large enough to ignore the transverse water recharge, and as demonstrated by Mao et al. [26], moisture movement in vadose zone is mainly triggered by evaporation, thus 1-D modelling is close to 3-D modelling if the main concern is on the vadose zone. Also please try to divide this sentence in to two.
Re: Thanks, we have corrected it. Please see line 300 to line 310.
- Line 234, add space between Figure and 5.
Re: Thanks, we have corrected it. Please see the caption of Figure 7 in the revised version.
- Line 309, update Figure .7 to Figure 7.
Re: Thanks, we have corrected it. Please see Figure 3 in the revised version.
- Line 326, Add “was”: The largest drop was
Re: Thanks, we have corrected it. Please see line 407.
- Line 327, Delete repeated word: 4.2. Groundwater Groundwater Level and Vadose Zone Moisture Change during the Monitoring Period
Re: Thanks, we have corrected it. Please see line 376.
- Please modify Table 2 to have 2017.9.13 in one line.
Re: Thanks, we have corrected it. Please see Table 3 in the revised version.
- Line 337: Add a space between Table 2 and line 337.
Re: Thanks, we have corrected it. Please see line 417.
- Line 337: Write Figure 8 instead of Figs. 8
Re: Thanks, we have corrected it. Please see 419.
- Authors are required to provide better comparison between the results from Figure 8 to point out the differences between different conditions and the reason for that.
Re: Thanks, we have explained the reason, please see line 427 to line 432, and line 464 to line 467.
- Line 344: add space between (a) and Sep. 13 and also add (d) for Mar. 1, 2018.
Re: Thanks, we have corrected it. Please see the caption of Figure 9 in line 415.
- From line 345 to conclusion: a series of discussion has been provided without explaining where they belong. Are they the discussion for the total project? Please specify.
Re: Thanks, we have added a separated section on discussion for the total project, please see section 7 in the revised version.
- Lines 346-348: please rewrite the sentence.
Re: Thanks, we have modified this sentence, please see line 418-419.
- Line 368: Update cm3/cm3 to cm3/cm3 to keep consistency in the manuscript
Re: Thanks, we have corrected it. Please see line 455.
- Line 391: please add “a” to The drop in groundwater level has a little effect on the moisture content of the vadose zone at depths of 16 m and above. Same comment for line 404.
Re: Thanks, we have corrected it. Please see line 512 and line 526.
- Is there any verification for the numerical model? Have the authors compared the results from numerical simulation with the in-situ results? The study focus on an important area but there is a lack of connection between different sections which required to be improved.
Re: Thanks, it is a good suggestion but we did not get all the data at present. Frankly, the in-situ tests were conducted by the open-pit mine corporation. Despite of this, the numerical simulation is based on the real data and parameters are determined based on in-situ tests by the authors. We will do further study on your suggestions, thanks again for your good comments.
We have modified the sections, the study methodology, results analysis and discussion are divided into separated section.
- Is there any recommendation to be provided by the authors? Did the model work successfully?
Re: Thanks, the study work and results in this paper guided the related government in Baorixile to supervise the dewatering processes in coal mining, in addition, the critical depth and the divided zone of vadose helps the ecological recovering works in the studied area, such as soil modification and revegetation.
- The conclusion needs to be improved as it only covers the summary of findings without provide a proper conclusion.
Re: Thanks, we have improved the conclusion. Additional discussion on the relationship between moisture in vadose zone and plants has been clarified in the conclusions.
Reviewer 5 Report
This is an interesting article which aims at evaluating the impact of coal mining operation on moisture movement in the vadose zone. The study evaluates an important aspect of the ecological system which is the soil moisture in a stressed environment. The study presents sound replicable methods and have provided empirical proof to the hypothesis presented.
The conclusions drawn from the study are sound and are based on evidence based results. However, as the authors mention that the results of this study can be used for formulating sustainable mining policies. Considering the scope of the journal, the authors should highlight how the results of this study can be used in developing these sustainable mining policies. Especially related to the importance of soil moisture, probably in terms of future mine closure or rehabilitation. The rehabilitation portion is just an idea that the author could use.
Author Response
Firstly, thanks greatly for your great comments and suggestions on our paper manuscripts, which improved the quality of our paper to a large extent. We accept all your opinions and the corrected points have been marked by red color or track formation.
Thanks again for your great efforts on prompting our paper.
Reply to #REVIEWER 5
This is an interesting article which aims at evaluating the impact of coal mining operation on moisture movement in the vadose zone. The study evaluates an important aspect of the ecological system which is the soil moisture in a stressed environment. The study presents sound replicable methods and have provided empirical proof to the hypothesis presented.
Re: Thanks greatly for your comments on improving our paper manuscripts.
The conclusions drawn from the study are sound and are based on evidence based results. However, as the authors mention that the results of this study can be used for formulating sustainable mining policies. Considering the scope of the journal, the authors should highlight how the results of this study can be used in developing these sustainable mining policies. Especially related to the importance of soil moisture, probably in terms of future mine closure or rehabilitation. The rehabilitation portion is just an idea that the author could use.
Re: Thanks, it is really a good suggestion. We have highlight this in the end of the introduction. Additionally, we have discussed on our study results on the ecological issues such as plants in the study area, which enables the related government and corporation to take right measurement on protecting the ecological system and make proper way to use the coal resources, governing the groundwater level to a suitable level, and design proper method to rehabilitation.
Round 2
Reviewer 1 Report
Congratulations on the refurbishment of the manuscript and thank you for addressing all the issues that I raised.
I have no further questions, good work!